# Ferroptosis-Mediated Cell Death Induced by NCX4040, The Non-Steroidal Nitric Oxide Donor, in Human Colorectal Cancer Cells: Implications in Therapy

**DOI:** 10.3390/cells12121626

**Published:** 2023-06-14

**Authors:** Birandra K. Sinha, Carl D. Bortner, Alan K. Jarmusch, Erik J. Tokar, Carri Murphy, Xian Wu, Heather Winter, Ronald E. Cannon

**Affiliations:** 1Mechanistic Toxicology Branch, Division of Translational Toxicology, National Institute of Environmental Health Sciences, Research Triangle Park, Durham, NC 27709, USA; erik.tokar@nih.gov (E.J.T.); carri.murphy@nih.gov (C.M.); xian.wu@nih.gov (X.W.); cannon1@niehs.nih.gov (R.E.C.); 2Laboratory of Signal Transduction, National Institute of Environmental Health Sciences, Research Triangle Park, Durham, NC 27709, USA; bortner@niehs.nih.gov; 3Immunity, Inflammation, and Disease Laboratory, National Institute of Environmental Health Sciences, Research Triangle Park, Durham, NC 27709, USA; alan.jarmusch@nih.gov (A.K.J.); heather.winter@nih.gov (H.W.)

**Keywords:** ferroptosis, NCX4040, erastin, ferrostatin-1, RSL3, colon cancer

## Abstract

Our recent studies show that the treatment of human ovarian tumor cells with NCX4040 results in significant depletions of cellular glutathione, the formation of reactive oxygen/nitrogen species and cell death. NCX4040 is also cytotoxic to several human colorectal cancer (CRC) cells in vitro and in vivo. Here, we examined the ferroptosis-dependent mechanism(s) of cytotoxicity of NCX4040 in HT-29 and K-RAS mutant HCT 116 colon cell lines. Ferroptosis is characterized by the accumulation of reactive oxygen species (ROS) within the cell, leading to an iron-dependent oxidative stress-mediated cell death. However, its relevance in the mechanism of NCX4040 cytotoxicity in CRCs is not known. We found that NCX4040 generates ROS in CRC cells without any depletion of cellular GSH. Combinations of NCX4040 with erastin (ER) or RSL3 (RAS-selective lethal 3), known inducers of ferroptosis, enhanced CRC death. In contrast, ferrostatin-1, an inhibitor of ferroptosis, significantly inhibited NCX4040-induced cell death. Treatment of CRC cells with NCX4040 resulted in the induction of lipid peroxidation in a dose- and time-dependent manner. NCX4040 treatment induced several genes related to ferroptosis (e.g., CHAC1, GPX4 and NOX4) in both cell lines. Metabolomic studies also indicated significant increases in both lipid and energy metabolism following the drug treatment in HT-29 and HCT 116 cells. These observations strongly suggest that NCX4040 causes the ferroptosis-mediated cell death of CRC cells. Furthermore, combinations of NCX4040 and ER or RSL3 may contribute significantly to the treatment of CRC, including those that are difficult to treat due to the presence of Ras mutations in the clinic. NCX4040-induced ferroptosis may also be a dynamic form of cell death for the treatment of other cancers.

## 1. Introduction

Cancer is one of the leading causes of death worldwide. Although breast and lung cancers are the most prevalent, colorectal cancer (CRC) represents third in morbidity and second in mortality of the total reported cancer deaths [1,2]. The incidence of CRC appears to be on the rise [3]. Current treatments for CRC include surgery, chemotherapy, radiotherapy, immunotherapy and targeted therapy. Studies have shown that an early diagnosis of CRC significantly improves the survival rate of patients. Unfortunately, CRC is usually detected at the advanced stages, leaving chemotherapy the primary choice with poor survival [4]. This failure appears to result from the development of chemotherapy resistance with undesirable drug side effects. Therefore, the development of newer drugs, selectively effective against CRC, is urgently needed.

The non-steroidal nitric oxide donor NCX4040, originally synthesized as an anti-carcinogenic compound, is highly cytotoxic to various tumors, including human CRC and ovarian cells [5,6]. Because NCX4040 shows no significant toxicity in vivo [5], the development of analogs of NCX4040 is highly desirable. Therefore, understanding the precise molecular mechanisms of the cytotoxicity of NCX4040 is essential for the future development of more selective and effective NCX4040 analogs. Studies suggest that NCX4040 induces its cytotoxicity by releasing ^●^NO following hydrolysis by tumor esterases, depleting cellular glutathione and causing oxidative stress [7,8,9]. Our recent studies have also shown a significant depletion of GSH and the formation of ROS by NCX4040 in human ovarian tumor cells [6]. Furthermore, our studies indicate that NCX4040 treatment results in the induction of both the CHAC1 and NOX4 genes in ovarian tumor cells [6]. Recently, *CHAC1* and *NOX4*, along with *GPX4*, have been suggested to be hallmarks of ferroptosis in tumor cells [10].

Ferroptosis is a non-apoptotic iron-dependent form of cell death, resulting from the formation of ROS/RNS in cells followed by the induction of cellular lipid peroxidation and membrane damage [11,12]. Ferroptosis, therefore, results from oxidative damage caused by the formation of ROS/RNS and the presence of ferrous iron and H_2_O_2_ (the Fenton reaction), or lipid peroxidation mediated by iron-containing lipoxygenases [13,14,15]. Although free ferrous iron is not accumulated in cells, it can be generated by proteins involved in cellular iron metabolism, such as transferrin receptor 1, ferritin and ferroportin. The antioxidant defenses involving glutathione peroxidase 4 (GPX4), which utilizes glutathione as the cofactor, reduces hydroperoxide lipids, inhibiting ferroptosis-mediated cell death [16,17].

NCX4040 has been shown to induce apoptosis in a variety of tumor cells, resulting from the formation of both ROS and RNS [6,7,18]. However, so far, ferroptosis has not been investigated in the mechanism of NCX4040-dependent tumor cell death. As ferroptosis emerges as a promising approach for cancer therapy [19,20], and CRC tumor cells undergo facile ferroptosis [21], we examined the role of ferroptosis in the mechanism of NCX4040-induced cell death in human HT-29 and HCT 116 CRC cells. Studies presented here show that NCX4040 indeed induces ferroptosis in both HT-29 and HCT 116 tumor cells. Furthermore, combinations of NCX4040 with ER or RSL3 may provide a better treatment modality for the therapy of CRC in the clinic.

## 2. Methods and Materials

Materials: NCX4040 was purchased from Sigma Chemicals (St. Lois, MO, USA) and was dissolved in DMSO. Erastin (2-[1-[4-[2-(4-Chlorophenoxy)acetyl]-1-piperazinyl]ethyl]-3-(2-ethoxyphenyl)-4(3*H*)-quinazolinone), RSL3 and Ferrostatin-1 were purchased from Cayman Chemicals (Ann Arbor, MI, USA) and were dissolved in DMSO. Stock solutions were stored at −80 °C. Fresh drug solutions, prepared from the stock solutions, were used in all experiments. Antibodies to GPX4, CHAC1, NOX4 and β-actin were purchased from Abcam (Waltham, MA, USA). 

### 2.1. Cell Culture 

Authenticated human colon tumor cells, HT-29 cells and HCT 116 cells were obtained from ATCC (Manassas, VA, USA) and were grown in Phenol Red-free RPMI 1640 media supplemented with 10% fetal bovine serum and antibiotics. Tumor cells were routinely used for 20–25 passages, after which the cells were discarded, and a new cell culture was started from the frozen stock.

### 2.2. Cytotoxicity Studies

The cytotoxicity studies were carried out with both a cell growth inhibition assay and Trypan Exclusion methods. Briefly, 50,000–60,000 cells/well were seeded onto a 6-well plate (in duplicate) and allowed to attach for 18 h. Various concentrations of drugs (NCX4040 or combinations of NCX4040), and minimally cytotoxic concentrations of ER, ferrostatin-1 (FeS) or RSL3 were added to cells (HT-29 or HCT 116) in fresh complete media (2 ML) and were incubated for 24, 48 or 72 h. When used, ER, FeS or RSL3 were preincubated with cells for 1–2 h before the addition of NCX4040. DMSO (0.01–0.1%) was included as the vehicle control when used. Following trypsinization, surviving cells were collected and counted in a cell counter (Beckman, Brea, CA, USA). For the trypan blue exclusion assay, 15 µL of cell mixtures was combined with 15 µL of trypan blue and counted in a T20 automatic cell counter (Bio-Rad, Hercules, CA, USA).

### 2.3. Flow Cytometric Analysis of Mitochondrial ROS

The analysis of mitochondrial ROS was determined by loading the cells with MitoSox Red (5 uM final concentration; Life Technologies, Carlsbad, CA, USA) for 30 min at 37 °C with a 7% CO_2_ atmosphere before the addition of the drug. Cells were examined at 2 h intervals with the addition of Sytox Blue as a vital dye via flow cytometry. An LSRFortessa flow cytometer (Benton Dickinson, San Jose, CA, USA), equipped with FACSDiVa software, was used to analyze all samples. MitoSox and Sytox Blue were excited using a 561 nm and 405 nm laser and detected using a 610/20 nm and 450/50 nm filter, respectively. For each sample, 10,000 cells were analyzed using FACSDiVa software.

### 2.4. Flow Cytometric Analysis for Intracellular Glutathione

Intracellular glutathione was determined as previously described [6]. Briefly, monochlorobimane dye (mBCl,10 µM final concentration; Life Technologies, Carlsbad, CA, USA) was added to each sample for 15 min at 37 °C with a 5% CO_2_ atmosphere prior to examination. Propidium iodide (PI) was added (final concentration of 5 ug/mL) to the samples before flow cytometric analysis using an LSRFortessa flow cytometer (Benton Dickinson, San Jose, CA, USA) equipped with FACSDiVa software. mBCl and PI were excited using a 405 nm and 561 nm laser and detected using a 530/30 nm and 582/15 nm filter, respectively. For each sample, 10,000 cells were analyzed using FACSDiVa software.

### 2.5. Lipid Peroxidation Assay

The assay for the peroxidation of cellular lipids was carried out by measuring the formation of malondialdehyde (MDA) using 2-thiobarbituric acid as previously published [22,23]. Briefly, about 2–3 × 10^6^ cells (HT-29 or HCT 116) were incubated with various concentrations of NCX4040 for 2–4 h at 37 °C. Following incubation, the reactions were stopped by adding 2% trichloroacetic acid, and the mixtures were centrifuged (5 min at 1000 *g*). Aliquots (1.0 mL) of the supernatant fractions were then reacted with 2.0 mL of 1% 2-thiobarbituric acid, and the chromophore was developed at 90 °C for 10 min. After the samples were cooled, the absorbance at 532 nm was determined.

### 2.6. Real Time RT-PCR

The expression levels of selected transcripts were examined via a real-time polymerase chain reaction (RT-PCR) using absolute SYBR green ROX Mix (ThermoFisher Scientific, Rochester, NY, USA) as previously described [6]. Total RNA was isolated using Trizol following treatment with NCX4040 (5 µM) for 4 and 24 h and was purified. Data were analyzed using the ΔΔCt method of relative quantification, in which cycle times were normalized to β-actin (or GADPH) from the same sample. Primers for the selected genes were designed using Primer Express 1.0 software and, in some cases, were synthesized (Integrated DNA technologies, CA, USA) from the published literature or were purchased from Origene (Gaithersburg, MD, USA). All real-time fluorescence detection was carried out on an iCycler (Bio-Rad, Hercules, CA, USA). Experiments were carried out three different times, and the results are expressed as the mean ± SEM. Analyses were performed using an unpaired Student’s *t*-test and considered significant when *p* ≤ 0.05.

### 2.7. Western Blot Assay

Western Blot analyses for various proteins were carried out following treatment with NCX4040 (5 µM) for 4 and 24 h using standard methods. Samples (5–20 μg of total protein) were electrophoresed under reducing conditions on 4–8% Tris-acetate gels (Novex, Life Technologies, Carlsbad, CA, USA) for 50 min at 200 volts. After electrophoresis, proteins were transferred onto nitrocellulose membranes and probed with anti-NOX4, CHAC1, GPx4 and anti-β-actin antibodies. An Odyssey infrared imaging system (Li-Cor Biosciences, Lincoln, NE, USA) was used to acquire images.

### 2.8. Metabolomics Studies

Cell culture extracts were analyzed using untargeted metabolomics, specifically ultra-high performance liquid chromatography (UHPLC) and high-resolution tandem mass spectrometry (HRMS/MS). MS data were acquired from samples (*n* = 1 injections) for comparison and statistical analysis. MS/MS data, used in the annotation of untargeted metabolomics features, were acquired from a pooled quality control sample analyzed multiple times via the AcquireX deep scan approach. Compound Discoverer 3.3.0.550 was used to process raw files to provide a tabulated output of features (i.e., unique descriptors of *m*/*z* and retention time) and corresponding annotations (MS/MS database matching). Outputs were formatted and further processed using in-house R scripts (via Jupyter Notebooks) to clean data prior to statistical analysis. Additional details are available in the Appendix A. The data supporting the annotated chemicals displayed in the figures were manually reviewed for MS, MS/MS and retention time matching (when available) (see Appendix A).

### 2.9. Statistical Analysis

The results are expressed as mean ± SEM of a minimum of 3 independent experiments (*n* = 3). A one-way analysis of variance (ANOVA) was used for statistical analysis using Graph Pad Prism (GraphPad Software, Inc., La Jolla, CA, USA). For multiple comparisons, Tukey’s multiple comparisons test was utilized, considering statistical significance when *p* < 0.05.

## 3. Results

### 3.1. Cytotoxicity Studies with NCX4040

Our previous studies have shown that NCX4040 is cytotoxic to human ovarian tumor cells [6]. In this study, we found NCX4040 to be significantly cytotoxic to both human colon HT-29 and HCT 116 cells (Figure 1A,B), and there were no significant differences in cytotoxicity, as the IC_50_ values were very similar (Figure 1C).

### 3.2. Measurements of Mitochondrial ROS

Because NCX4040 generates ROS in various tumor cells, which leads to tumor cell death, we investigated whether NCX4040 also forms ROS in these colon cancer cells. We used Mitosox for the detection of mitochondrial ROS as previously described [6]. Although the use of Mitosox has remained controversial [24], we [6] as well as others [7,25] have utilized Mitosox successfully to detect ROS in cells. Our results (Figure 2) clearly show that NCX4040 generates ROS in a dose-dependent manner in both cells. At higher concentrations of NCX4040 (e.g., 10 µM), more ROS were detected in HCT 116 cells than in HT-29 cells.

### 3.3. Lipid Peroxidation in HT-29 and HCT 116 Cells

Examination for the peroxidation of HT-29 and HCT cellular lipids indicated that MDA formation increased in both cells over the controls in the presence of NCX4040 in time- and dose-dependent manners. Although increases in MDA formation were small, these increases were statistically significant (Figure 3).

### 3.4. RT-PCR Studies

#### 3.4.1. NCX4040 Induces Oxidative Stress Genes in HT-29 and HCT 116 Cells

Our previous studies have shown that NCX4040 treatment results in modulations of various genes related to oxidative stress, inflammation and DNA responses in human ovarian tumor cells [6,26]. Therefore, we evaluated oxidative-stress-related genes in HT-29 and HCT 116 cells following NCX4040 treatment. Studies show that oxidative stress genes are significantly modulated by NCX4040 (Figure 4A,B). NCX4040 treatment resulted in a rapid (4 h) induction of the HMOX1/OX1 gene (6-fold) in HT-29 cells and more than 12-fold in HCT116 cells. The HMOX1/OX1 gene remained elevated at 24 h in HCT 116 cells (3-fold); however, it decreased to the control values in HT-29 cells. The HMOX1/OX1 gene is considered to be an important biomarker of oxidative stress and is induced by a number of free-radical-producing drugs and chemicals [27,28]. Ferroptosis markers, *GPX4*, *NOX4* and *CHAC1* were also significantly induced by NCX4040 in both cell lines at 24 h. However, *CHAC1*, which causes the cleavage of glutathione, resulting in the depletion of GSH [29], was rapidly induced (2–3-fold) in both HT-29 and HCT 116 cells at 4 h and remained elevated in both cells at 24 h, with 5–6 fold in HT-29 and 35–40-fold in HCT 116 cells. SOD_2_, which catalyzes the decomposition of superoxide radical anion (O_2_^●-^) formed from NOX4, was induced by NCX4040 only in HCT 116 cells. Furthermore, *OGG1*, which is responsible for the repair of oxidative damage in DNA caused by ROS, was elevated in both HT-29 and HCT 116 cells at 24 h (Figure 4).

#### 3.4.2. NCX4040 Induces Inflammatory Response Genes in HT-29 and HCT 116 Cells

The NCX4040 treatment of HT-29 and HCT 116 colon cells also induced various inflammatory response genes (Figure 5A,B), as we found previously in ovarian tumor cells [6]. NCX4040 significantly induced *VEGF* in HCT 116 cells at both 4 and 24 h, whereas it was only induced in HT-29 cells at 24 h. *COX2* was significantly induced in HCT116 cells, whereas NCX4040 had only a small effect in HT29 cells. It is also interesting that NCX4040 significantly decreased the expression of *NF-kB* in HT-29 at 4 h, whereas it had no effect in HCT 116 cells (Figure 5A).

### 3.5. Ferrostatin-1 Inhibits NCX4040 Cytotoxicity in HT-29 and HCT 116 Cells

Our RT-PCR results indicate the significant modulation of GPX4, NOX4 and CHAC1 genes by NCX4040 in HT-29 and HCT 116 cells. Because these genes are indicators of ferroptosis, we examined the effects of Ferrostatin-1 (FES) on the cytotoxicity of NCX4040 in both HT-29 and HCT 116 cells. FES is a known inhibitor of ferroptosis in tumor cells [30,31,32]. The results shown in Figure 6A,B indicate that FES (2 µM, minimally cytotoxic dose) significantly inhibited the cytotoxicity of NCX4040 in both HT-29 and HCT 116 cells, suggesting that NCX4040 induces ferroptosis in these CRC tumor cells. Furthermore, FES was more effective in inhibiting NCX4040-induced cytotoxicity in HCT 116 cells than in HT-29 cells (Figure 6B; compare 10^−5^ M induced cell killing).

### 3.6. Erastin Enhances NCX4040 Cytotoxicity in HT-29 and HCT 116 Cells

Erastin (ER) is a well-known inducer of ferroptosis in tumor cells and has been utilized extensively to decipher the mechanism of ferroptosis [33,34,35]. We used various concentrations of ER to examine its effects on NCX4040 cytotoxicity in CRC cells. Our results, shown in Figure 7A,B, clearly indicate that ER was effective in enhancing NCX4040-mediating CRC cell death. Again, HCT 116 cells were more sensitive to the ER-dependent enhancement of NCX4040 cytotoxicity (Figure 7B).

### 3.7. RSL3 Enhances NCX4040 Cytotoxicity in HT-29 and HCT 116 Cells

RSL3 (RAS-selective lethal 3) is a ferroptosis-triggering agent that has been utilized extensively to induce ferroptosis in various tumor cells [17,36,37]. Therefore, the effects of RSL3 on NCX4040 cytotoxicity were investigated, and we found that it significantly enhanced NCX4040 cytotoxicity in both cells (Figure 8A,B). Again, HCT 116 cells were found to be more sensitive to RSL3-induced NCX4040-mediated cytotoxicity.

### 3.8. Metabolomic Studies in HT-29 and HCT 116 Cells

In order to further understand the mechanisms of NCX4040-induced ferroptosis in CRC cells, we used untargeted metabolomics and examined the effects of NCX4040 on cellular glutathione levels, lipid metabolism and differential energy metabolism in HT-29 and HCT 116 cells. 

#### 3.8.1. NCX4040 Increases Glutathione in HT-29 and HCT 116 Cells

Our metabolomic studies show significant increases in glutathione (GSH) following the NCX4040 treatment of CRC cells (Figure 9). GSH levels were elevated at 24 h in both cells (Figure 9). The ratio of GSH to glutathione disulfide (GSSG), the oxidized form, remained similarly indicative of the cellular ability to maintain redox potential. Further, taurine, a reported antioxidant and anti-inflammatory mediator, was increased after 24 h (Figure 9C), indicating NCX4040-induced oxidative stress in CRC cells.

#### 3.8.2. NCX4040 Enhances Lipid Metabolism in HT-29 and HCT116 Cells

Lipid metabolism is a complex biochemical process that is involved in the regulation of cell survival and death, including ferroptosis [38]. Metabolomic studies indicated that NCX4040 treatment significantly increased lipid metabolism in these CRC cells (Figure 10). Acylcarnitines were observed to be statistically different between the control and 24 h samples in both cells. The acyl chain composition is reflective of energy metabolism patterns, including fatty acid oxidation, which may result in the formation of lipid radicals during beta oxidation, leading to increased cells death via ferroptosis.

The short-chain acylcarnitines (Figure 10A–C) increased after 24 h in HT-29 and HCT 116 cells. Increased levels suggest that energy production from glucose, amino acids or fatty acid degradation is increased after treatment. The dicarboxylic acid conjugate, succinyl-carnitine (Figure 10D), reflects the selective patterns observed in carnitines. In contrast, the long-chain acylcarnitines (Figure 10E–H) decreased in the 24 h samples. The observed decrease may reflect alterations in fatty acid metabolism, as the mitochondria are the primary location of synthesis and metabolism of long-chain acylcarnitines.

Metabolomics also indicated differences in arachidonic acid (AA) metabolism in these cells (Figure 11). AA is involved in prostaglandin synthesis and is also an important ingredient for ferroptosis [39].

#### 3.8.3. NCX4040 Enhances Energy Metabolism in HT-29 and HCT 116 Cells

Metabolomic studies also show significant increases in ATP and cellular respiration co-factors, NAD+ and FAD+ (Figure 12), in both cells at 24 h. This observation suggests increases in energy production related to cell survival. The observed increase in energy production may reflect increased DNA/RNA repair of oxidative damage induced by NCX404, as supported by the observed increase in GTP, a building block of RNA and DNA (Figure 12). 

## 4. Discussion

Ferroptosis is induced via iron-dependent lipid peroxidation following the cellular formation of ROS [11,12]. Although the mechanism of ferroptosis is under active investigation, it is different from those of other cell death processes, such as necrosis, autophagy, and apoptosis. The cellular death resulting from ferroptosis has been shown to arise from the inhibition of glutathione peroxidase 4 (GPX4) and the accumulation of intracellular lipid hydroperoxides (LOOH), resulting in damage to cellular membranes (lipid peroxidation) in the presence of iron [11,12,16]. The damaging species is the reactive ^●^OH, formed from the reaction of H_2_O_2_ with Fe^2+^ (the Fenton reaction). Several small molecules, e.g., ER and RSL3, have been reported to induce ferroptosis in tumor cells [33,34,36,37]. Several pathways have now been reported for ER-induced ferroptosis, including the inhibition of the system X_C_^–^ (glutamate/cystine antiporter) [35], the inhibition of the mitochondria-bound voltage-dependent anion channel (VDAC) [33] and the modulation of the tumor suppressor p53 gene [40,41]. System X_C_^–^ is a transmembrane cystine–glutamate antiporter that specifically imports extracellular L-cystine into cells in exchange for glutamate. Cystine, the disulfide form of cysteine, is imported into the cell by system X_C_^–^ and is reduced to cysteine, which is the key intermediate for the synthesis of glutathione (GSH), an important cellular antioxidant. Thus, the inhibition of the system X_C_^–^ by ER results in the depletion of the cellular GSH, leading to oxidative stress and ferroptosis-mediated cell death [12,42]. RSL3 is a potent ferroptosis-triggering agent that inhibits GPX4, thereby promoting ferroptosis, including in CRC cells [36]. RSL3 can kill RAS mutant cancer cells and activate the iron-dependent, nonapoptotic cell death of RAS mutant cancer cells [43]. Increases in GPX4 in tumor cells have been shown to inhibit ferroptosis [16,44].

However, ferroptosis has not been investigated in the treatment of CRC in the clinic. Therefore, the exploitation of ferroptosis for killing CRC cells in response to NCX4040 needs elucidation for therapy. Herein, we investigated the role of ferroptosis in the mechanism of NCX4040-dependent cell death in HT-29 and HCT 116, Ras mutated (codon 13) colon tumor cells. We utilized various inducers and inhibitors of ferroptosis to decipher the mechanisms of cell death. Our study shows that NCX4040 was equally cytotoxic to both CRC cells. Previous studies have shown that NCX4040 induces the significant depletion of cellular GSH and generates ROS formation in tumor cells [6,7]. In this study, we found that NCX4040 also generated ROS in HT-29 and HCT 116 cells in both a time- and dose-dependent manner. Fewer ROS were formed and detected at 2 h than at 4 h of drug treatment (not shown). Our studies also show that NCX4040 significantly increased lipid peroxidation in both HT-29 and HCT 116 cells in a time- and dose-dependent manner. These observations suggest that NCX4040-generated ROS must have reacted with cellular lipids to form lipid peroxides in these tumor cells.

It should be noted that both the formation of ROS and increases in lipid peroxidation were independent of cellular GSH, as we found no significant depletion of GSH in HT-29 and HCT 116 cells by our mCBI flow cytometric detection method. In contrast, the metabolomic studies clearly indicate that the treatment of CRC cells with NCX4040 resulted in increases in GSH following the 24 h drug treatment (Figure 11). We found that treatment with ER also failed to deplete GSH in these cells. These observations suggest that neither NCX4040 nor ER inhibits the Xc-anti-transporter in HT-29 and HCT 116 cells, indicating that other biochemical and metabolic changes besides the cellular depletion of GSH are responsible for the induction of ferroptosis by NCX4040 in CRC cells.

The examination of the effects of ER on the cytotoxicity of NCX4040 in HT-29 and HCT116 cells shows that ER was highly effective in enhancing NCX4040-dependent cell death in both CRC cells. More interestingly, we found that HCT 116 cells were more sensitive to combinations of NCX4040 and ER (Figure 7). Similarly, RSL3 also significantly enhanced NCX4040-mediated cell death in both HT-29 and HCT 116 cells. Again, HCT 116 cells were more sensitive to RSL3-NCX4040 combinations than HT-29 cells (Figure-8). In contrast, FES, a strong inhibitor of ferroptosis, significantly inhibited NCX4040-induced cell death in both HT-29 and HCT 116 tumor cells (Figure 6). Again, we found that FES was more effective in inhibiting NCX4040-dependent cell killing in HCT 116 cells. These observations, e.g., the inhibition of NCX4040-dependent killing by FES and the enhanced or synergistic killing of HT-29 and HCT 116 cells by ER-NCX4040 and RSL3-NCX4040 combinations, clearly suggest that NCX4040 induces ferroptosis in HT-29 and HCT 116 cells. Furthermore, HCT 116 cells were more sensitive to ferroptosis-dependent cell death than HT-29 cells. Although the reason for this is not clear, this may be due to the fact HCT116 cells contain a KRAS-mutation, and KRAS-mutant cells are more sensitive to both ER- and RSL3-induced ferroptosis [45]. Furthermore, HCT 116 cells contain WTp53, whereas HT-29 cells contain mutantp53. Ferroptosis has been suggested to require WTp53 activity [41]. In addition, recent reports suggest the degree of FAM193A expression is a widespread enabler of p53 activity in cell death [46], which may relate to the differential ferroptosis sensitivity of HT-29 and HCT 116.

Although the mechanisms of NCX4040-induced ferroptosis are not completely clear, some reasonable conclusions can be drawn from our studies. First, NCX4040-induced ferroptosis-mediated tumor cell death is ROS-dependent, as there was a significant increase in ROS formation in both HT-29 and HCT 116 cells. The formation of ROS in these cells is also confirmed by our RT-PCR studies, showing significant increases in *HMOX1/OX1*, a hallmark of oxidative stress in cells. Furthermore, both *SOD_2_* and *NOX4* were induced, indicating that the superoxide anion radical (O_2_^−●^) is formed in these cells. Our studies also show that lipid peroxides are generated in these cells, as lipid peroxidation was enhanced in the presence of NCX4040. Lipid peroxidation has been suggested to trigger ferroptosis. Lipid peroxides/lipid hydroperoxides (L-OOH) are known to cause damage to the cellular plasma membrane due to the accelerated oxidation of the membrane lipids. Furthermore, increases in the concentrations of lipid peroxides can induce damage to nucleic acids and proteins from toxic aldehydes formed from the oxidation of lipids, causing additional toxicity and inducing cell death by ferroptosis [15].

Our studies show that the combined treatment of NCX4040 and RSL3 increased the death of both HT-29 and HCT 116 CRC cells, due to the accumulation of cytotoxic lipid peroxidation products. RSL3, an inhibitor of GPX4 [16,17,36], enhanced cell death, which further confirms that lipid peroxides are responsible for cell death in HT-29 and HCT 116 cells. Our study also shows that GPX4 transcripts were significantly enhanced in both HT-29 and HCT 116 cells. GPX4 utilized GSH as a cofactor to eliminate hydroperoxides, and our metabolomic studies show that cellular GSH was increased by NCX4040 in these cells.

It should be noted that the CHAC1 gene (and proteins) was significantly enhanced in both cells, albeit significantly more in HCT 116 cells. *CHAC1*, in addition to *NOX4* and *GPX4*, is considered to be a hallmark of ferroptosis and has been reported to be involved in tumor cell death via the induction of ferroptosis [12,47,48,49]. Although CHAC1 is known to hydrolyze glutathione, leading to the depletion of GSH in tumor cells, our study does not show the depletion of GSH in either cell line. However, *CHAC1* is also known to induce ER stress via the *ATF4-CHOP-CHAC1* pathway, leading to ferroptosis [10,48].

Protein levels of ferroptosis-associated CHAC1, GPX4 and NOX4 genes were evaluated using the Western blot methods. The CHAC1 protein was induced following NCX4040 treatment in both cell lines, with significantly more in HCT 116 cells. Similarly, GPX4 proteins levels were also induced by NCX4040; however, protein levels of NOX4 remained unchanged in both HT-29 and HCT 116 cells. Although protein expressions did show increases following treatment with NCX4040, there was no accord between transcript expressions with protein expressions. This may be due to differences in the stability/half-life of transcripts compared to proteins. It is possible that the half-lives of proteins are shorter due to a rapid degradation or turnover. Furthermore, it is also possible that these proteins are S-nitrosylated by ^●^NO/^●^NO-derived species formed from NCX4040 in cells, resulting in post-translational modifications, including the decreased stability of proteins as reported by us as well as others [50,51,52].

Our studies also show that various inflammatory response genes, e.g., *COX2, VEGF* and *NF-kB*, were also modulated by NCX4040 in these CRC cells. These observations are consistent with our findings with NCX4040 in human ovarian cells [6]. Although the exact roles of these genes in NCX4040-induced ferroptosis is not known at this time, several studies indicate that the COX2 and VEGF genes are involved in the process of ferroptosis [53,54,55,56]. COX2 has been found to be increased during ferroptosis and has been suggested to be a biomarker, as inhibitors of the COX2 enzyme failed to modulate ferroptosis [57].

Finally, the inhibition of the mitochondria-bound voltage-dependent anion channel (VDAC) has been suggested to play an important role in the mechanism of ferroptosis. The impairment of mitochondrial functions can increase the sensitivity of anticancer drugs to cancer cells. The VDAC is an ion channel located in the outer mitochondrial membrane, where it mediates and controls molecular and ion exchange between the mitochondria and the cytoplasm. The permeability of the VDAC can be altered by drugs, causing mitochondrial metabolic dysfunction, ROS production, and oxidative stress-mediated death. Yagoda et al. [33] reported that ER changes the permeability of the mitochondrial outer membrane and that the VDAC is the target of ER. ER was shown to reverse tubulin’s inhibition of the VDAC in vitro and in vivo, allowing the VDAC to open [33]. Opening of the VDAC leads to various biological effects, including an increase in mitochondrial metabolism (the increase in Δψ), a decrease in glycolysis (anti-Warburg effect) and an increase in ROS production and oxidative stress [58]. The anti-Warburg action can lead to damage to cancer cells and decreases in cell proliferation. In addition, ER can hyperpolarize mitochondria in cancer cells, which is followed by rapid depolarization, resulting in mitochondrial dysfunction. ^●^NO, delivered via NO-donors, has been reported to directly inhibit VDAC functions [59]. Although the effects of NCX4040 on VDAC were not examined here, it is very likely that ^●^NO formed from NCX4040 inhibits/interferes with the VDAC functions of CRC cells, generating ROS (as observed here) without affecting cellular GSH status. Increased/synergistic CRC cell death via the combination of NCX4040 and ER may result from this combined effect of both agents on VDAC.

## 5. Conclusions

Studies presented here show that NCX4040 induces the formation of ROS in HT-29 and HCT 116 colon cancer cells without significantly affecting cellular GSH. The NCX4040 treatment of these tumor cells resulted in increases in lipid peroxidation. Combinations of ferroptosis inducers erastin and RSL3 significantly enhanced NCX4040 cytotoxicity, whereas ferrostatin-1, an inhibitor of ferroptosis, significantly inhibited NCX4040 cytotoxicity in CRC cells. Our studies also show that the treatment of HT-29 and HCT 116 cells resulted in significant modulations of CHAC1, GPX4, NOX4 and COX2 genes, biomarkers of ferroptosis. These events, taken together, strongly suggest that NCX4040 induces ferroptosis in CRC cells. Combinations of erastin or RSL3 with NCX4040 may provide a better treatment modality for CRC in the clinic.

## Figures and Tables

**Figure 1 cells-12-01626-f001:**
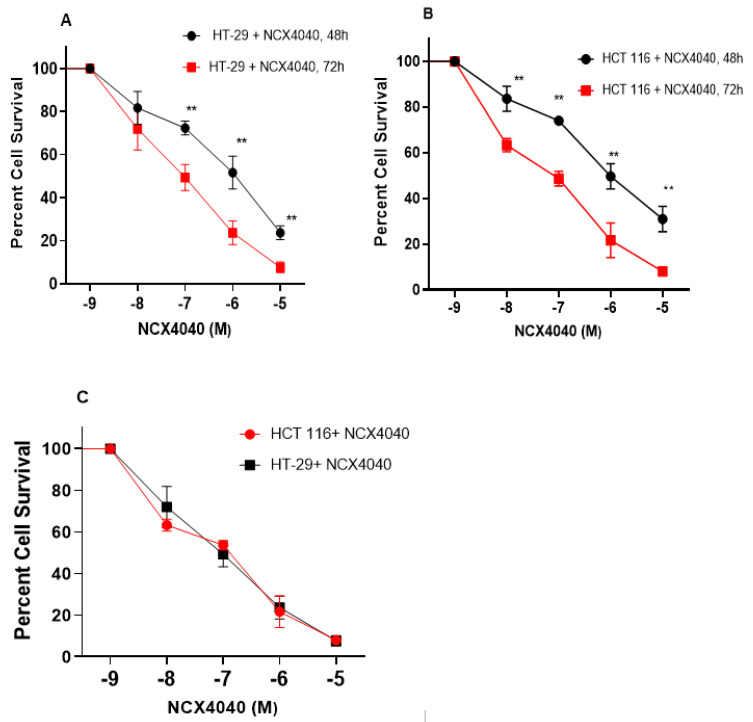
Cytotoxicity of NCX4040 in HT-29 (**A**) and HCT 116 (**B**) following 48 and 72 h drug exposure, respectively. Plot of cytotoxicity curves of NCX4040 in HT-29 and HCT 116 showing similar cytotoxicity (**C**). ** *p*-values < 0.005 compared to untreated control.

**Figure 2 cells-12-01626-f002:**
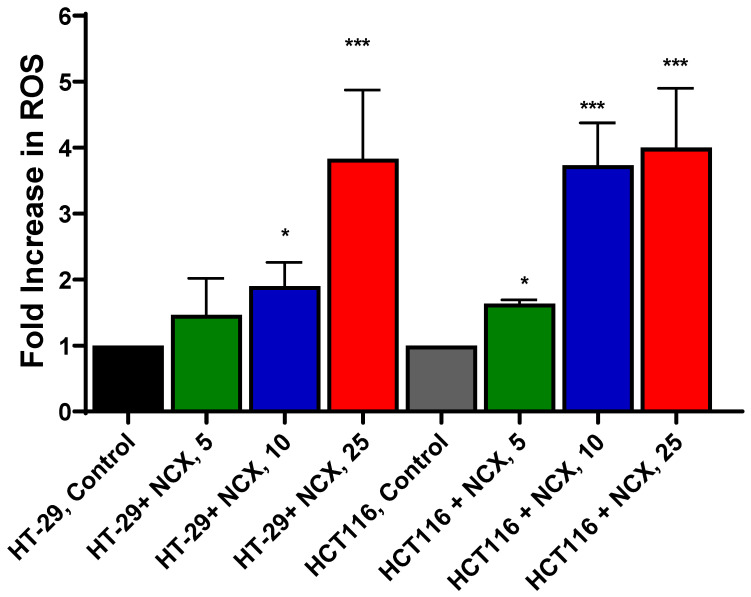
Dose-dependent ROS formation as detected by Mitosox from NCX4040 in HT-29 and HCT116 cells following a 4 h treatment. * and ***, *p*-values < 0.05 and 0.001, respectively, compared to untreated control.

**Figure 3 cells-12-01626-f003:**
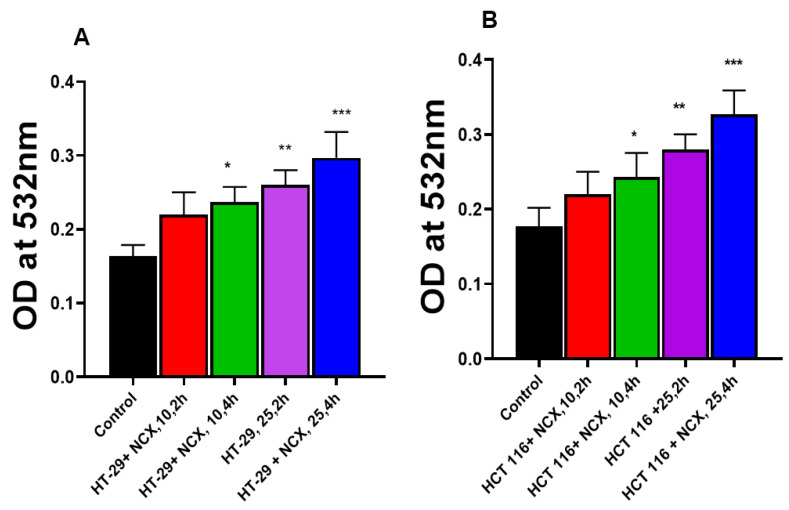
Dose and time dependence of NCX4040-induced lipid peroxidation in HT-29 (**A**) and HCT 116 (**B**). MDA formation was measured at 532mM. *, ** and ***, *p*-values < 0.05, 0.005 and 0.001, respectively, compared to untreated control.

**Figure 4 cells-12-01626-f004:**
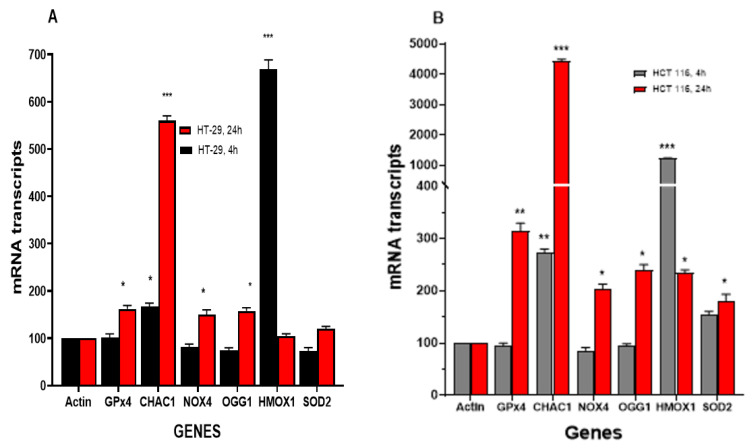
Effects of NCX4040 (5 µM) on various oxidative-stress-related genes in HT-29 (**A**) and HCT 116 (**B**) cells following treatment with NCX4040 for 4 and 24 h. *, ** and ***, *p*-values < 0.05, 0.005 and 0.001, respectively, compared to control (β-Actin at 4 h and 24 h, respectively).

**Figure 5 cells-12-01626-f005:**
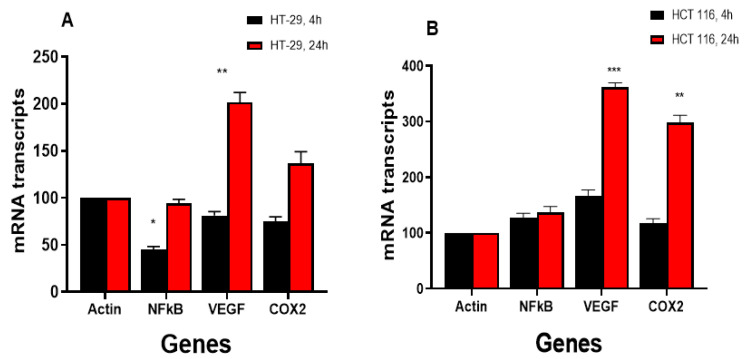
Effects of NCX4040 (5 µM) on various inflammatory response genes in HT-29 (**A**) and HCT 116 (**B**) cells following treatment with NCX4040 for 4 and 24 h. *, ** and ***, *p*-values < 0.05, 0.005 and 0.001, respectively, compared to control (β-Actin at 4 h and 24 h, respectively).

**Figure 6 cells-12-01626-f006:**
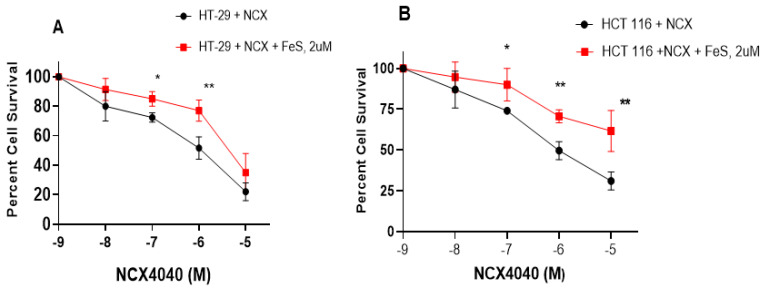
Effects of ferrostatin-1 (2 µM) on the cytotoxicity of NCX400 in HT-29 (**A**) and HCT 116 (**B**) cells following 48 h of drug treatment. FES was pre-incubated with the cells for 1–2 h before adding NCX4040. * and **, *p*-values < 0.05 and 0.005, respectively, compared to treatment matched samples.

**Figure 7 cells-12-01626-f007:**
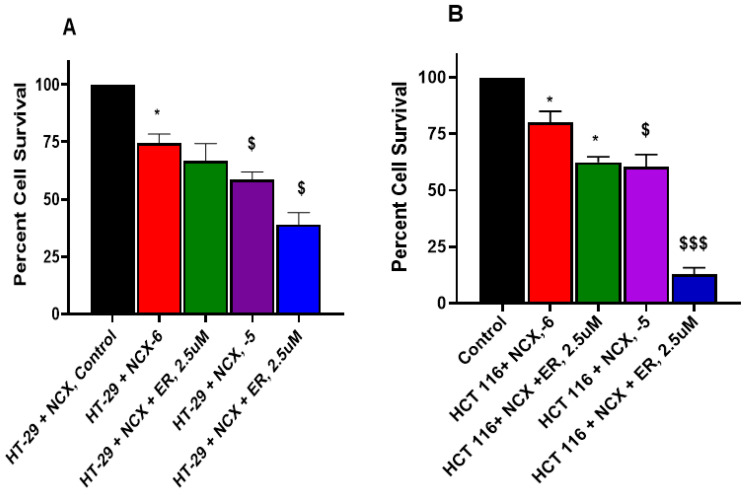
Erastin-dependent enhancement of NCX4040 (10^−6^ M and 10^−5^ M) cytotoxicity in HT-29 (**A**) and HCT 116 (**B**) tumor cells following 24 h drug treatments. ER was pre-incubated with cells for 1-2 h before adding NCX4040. * and $, *p*-values < 0.05 compared to 10^−6^ M NCX4040 and the control. $$$, *p*-values < 0.001 compared to 10^−5^ M NCX4040.

**Figure 8 cells-12-01626-f008:**
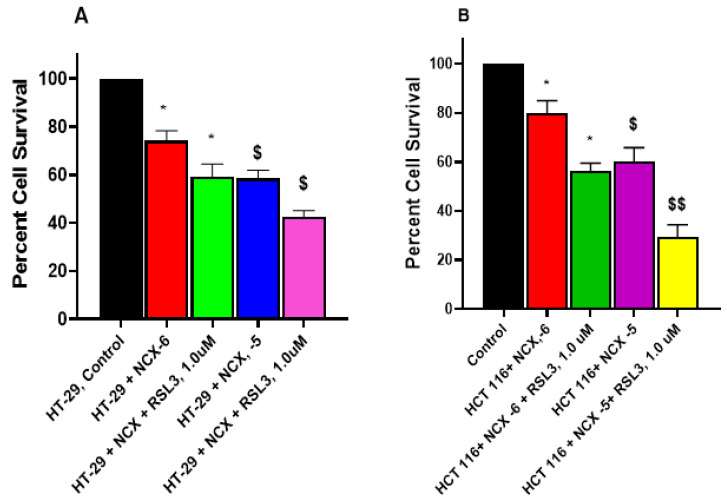
RSL3-induced enhancement of NCX4040 (10^−6^ M and 10^−5^ M) cytotoxicity in HT-29 (**A**) and HCT 116 (**B**) tumor cells following 24 h drug exposures. RSL3 was pre-incubated with cells for 1–2 h before adding NCX4040. * and $, *p*-values < 0.05 compared to the control and 10^−6^ M NCX4040. $$, *p*-values < 0.005 compared to 10^−5^ M NCX4040.

**Figure 9 cells-12-01626-f009:**
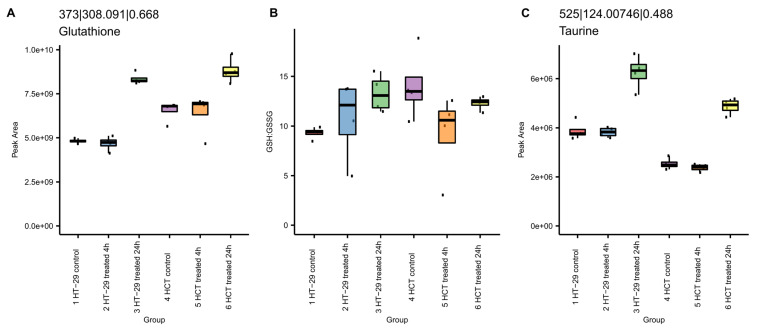
Effects of NCX4040 on GSH, GSH/GSSG and taurine in HT-29 and HCT 116 cells. Box plots display the (**A**) peak area of feature 373 (*m*/*z* 308.091 at 0.67 min) annotated by MS/MS as glutathione, the (**B**) computed peak area ratio of glutathione (GSH) to glutathione disulfide (GSSG), and the (**C**) peak area of feature 525 (*m*/*z* 124.0075 at 0.49 min) annotated by MS/MS as taurine. Boxes indicate first and third quartiles, and whiskers indicate 1.5 times the inter-quartile range. The median is displayed. Individual data points are indicated.

**Figure 10 cells-12-01626-f010:**
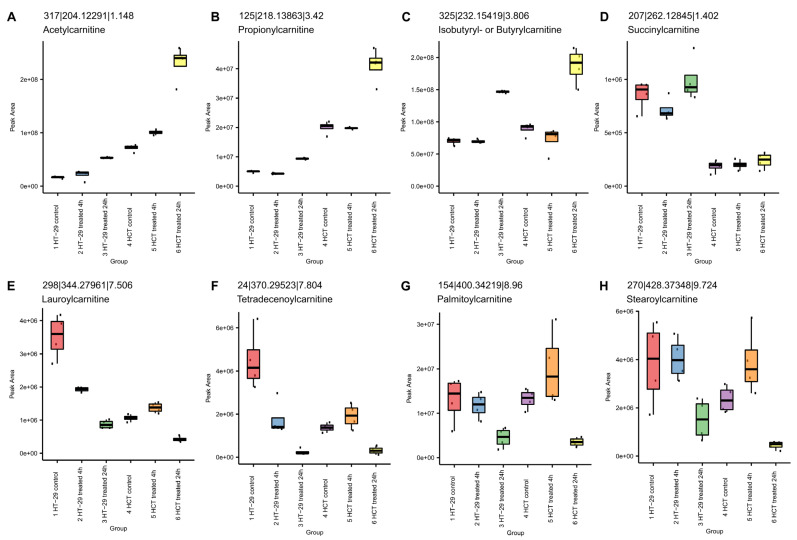
Effects of NCX4040 on the formation of acycarnitines in HT-29 and HCT 116 cells. Box plots display the peak area of features annotated by MS/MS as acyl carnitines in order of increasing chain length: (**A**) Acetylcarnitine, (**B**) Propionylcarnitine, (**C**) Isobutyrylcarnitine, (**D**) Succinylcarnitine, (**E**) Lauroylcarnitine, (**F**) Tetradecenoylcarnitine, (**G**) Palmitoylcarnitine and (**H**) Stearoylcarnitine. Boxes indicate first and third quartiles, and whiskers indicate 1.5 times the inter-quartile range. The median is displayed. Individual data points are indicated.

**Figure 11 cells-12-01626-f011:**
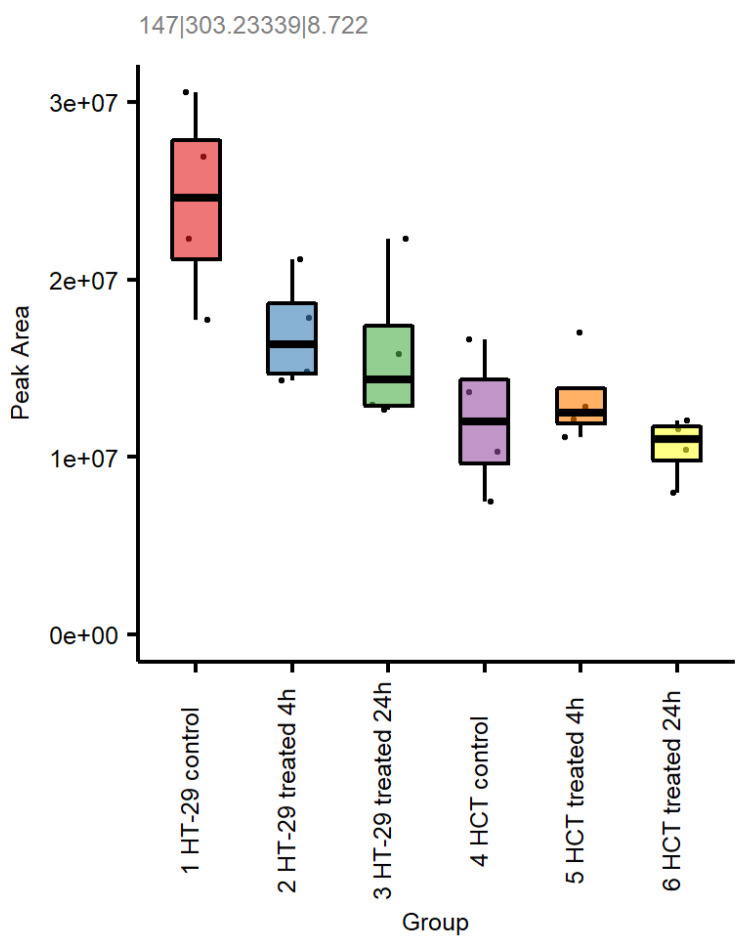
Effects of NCX4040 on arachidonic acid formation in HT-29 and HCT 116 cells. Feature 147|303.23339|8.722 (negative mode data) is putatively annotated as arachidonic acid.

**Figure 12 cells-12-01626-f012:**
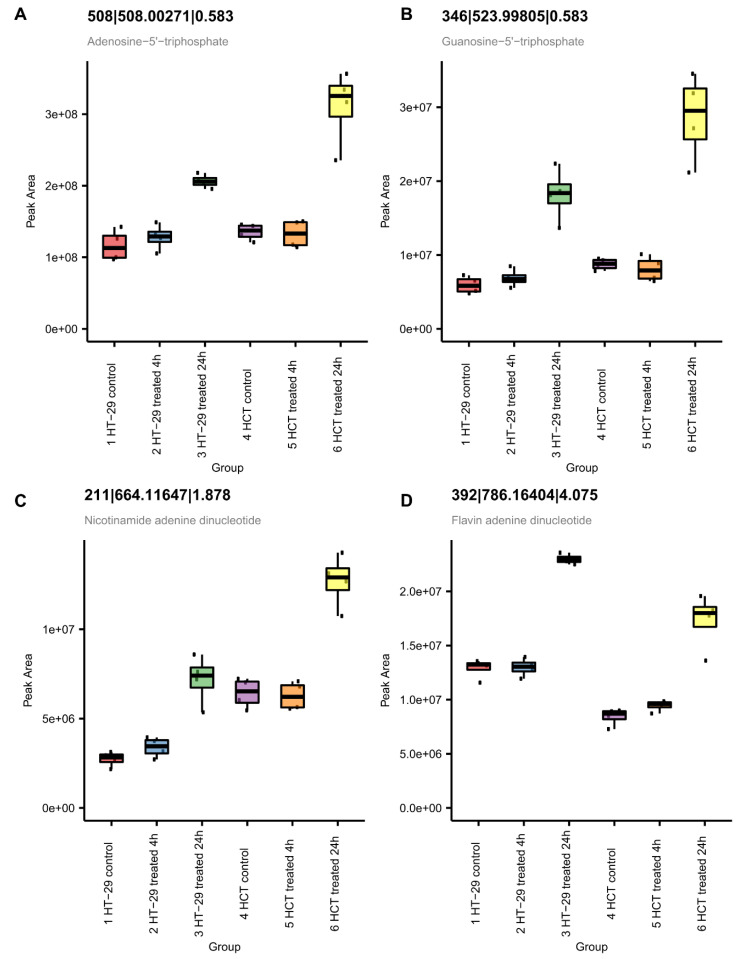
Metabolomic measurement of ATP (**A**), NAD+ (**B**), FAD+ (**C**) and GTP (**D**) in HT-29 and HCT 116 cells following NCX4040 treatment.

## Data Availability

Not applicable.

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
