# Peer review of "Ferroptosis-Mediated Cell Death Induced by NCX4040, The Non-Steroidal Nitric Oxide Donor, in Human Colorectal Cancer Cells: Implications in Therapy"

_cells, 2023, doi:10.3390/cells12121626_

Round 1
Reviewer 1 Report
Here the study aimed to investigate the involvement of ferroptosis in the mechanism of NCX4040-induced cell death in HT-29 and HCT116 CRC cells. Findings are convincing to demonstrate that NCX4040 indeed induces ferroptosis in both HT-29 and HCT116 tumor cells. Also, a strategy of combining NCX4040 with ER or RSL3 to represent a more effective treatment approach for CRC in clinical settings is interesting. However, a few additional questions have to be addressed to enhance the interest of the readers.
Major Comments.
1. Authors have tested for several oxidative stress genes in Fig4. It would be interesting any changes in NRF1, NRF2 and KEAP upon treatment
2. Cetuximab is the first line of treatment for CRC and also induces ROS and DNA damage. Does a combination of cetuximab with NCX4040 further enhances the synergy in cell killing? Does it have a distinct phenotype or molecular mechanisms?
Minor comments:
1. Authors should extensively work on the figures. For example, Figures 9 and 10 are difficult to understand the labeling.
Author Response
Reviewer-1
Major Comments
- In this study we did not check for expressions of NRF1 or 2 as our previous gene expression study with NCX4040 in ovarian cells did not show any differential expression of NRF’s. However, HMOX1 was highly expressed. It is possible that these genes may be affected by NCX4040 in CRC cells. We have recently initiated a large study on gene expression profiling in CRC cells, including HT-29, HCT116 and DLD-1 cells following treatment with NCX4040 and erastin.
- We are aware of Cetuximab-dependent ferroptosis in CRC (Referenced here-43). As mentioned in comment 1, our future study also includes examining interactions of adriamycin, cis-platinum and topotecan with NCX4040, erastin, and RSL3 in CRC cells (HT-29, HCT116 and DLD-1). Furthermore, we have also designed, based on in-silico modeling, new analogs of NCX4040 which are currently being synthesized for future studies.
Thank you very much for these suggestions which we will include in our future studies.
- Figures are now improved, especially 9 and 10.
Reviewer 2 Report
I have reviewed the manuscript entitled. Although it is an interesting topic, but a few points are suggested, needs few editing.
1. In the abstract section: NCX404 should be corrected.
2. In all parts: HCCT 116 should be corrected (written the same)
3. In section 3-2: The sentence "...At lower concentrations of" should be modified to” At higher concentration…”
4. In the section of gene expression: 24 hours’ drug treatment is mentioned, but in Figure 4: period of 4 and 24 hours is written to be corrected.
5. For a more accurate comparison: it would have been better to check the gene expression before and after the treatment.
6. Figure 9, 10 is not clear.
7. Check again in terms of writing.
8. Is IC fifty calculated? not mentioned.
9. On what basis is the drug concentration selected?
-
Author Response
Reviewer-2
1-3: We have corrected these in the revised manuscript.
- We carried out RT-PCR studies at both 4h and 24h (section 2.6) and mentioned in section 3.4. The figure showing data with 4 and 24h is correct.
- The gene expression at Zero (0) time (control) is before treatment and changes in gene expression at 4h and 24 h are compared to this control.
- We have improved them to made them clear. Figures are now improved, especially 9 and 10.
- All the corrections have been made.
- No need to show IC50 as there is no significant differences between cell lines as shown in Figure-1.
- Cytotoxicity studies were carried at 24, 48 and 72 h, respectively, with NCX4040 in cells. At 5 µM NCX4040, about 25% of cell death occurred in both cell lines at 24h drug treatment. We wanted to use a concentration of NCX4040 that was slightly cytotoxic (but not extremely cytotoxic) as this indicated genes that were affected by NCX4040 and possibly involved in cell death. Similarly, cytotoxicity studies were carried out with erastin, ferrostatin-1 and RSL3 in these cells and we chose minimally toxic doses for synergistic or antagonist interactions with NCX4040 in cells.